# Dual space learning with variational autoencoders

**Hirono Okamoto, Masahiro Suzuki, Itto Higuchi, Shohei Ohsawa, Yutaka Matsuo**
The University of Tokyo
{h-okamoto,masa,itto.higuchi,ohsawa,matsuo}@weblab.t.u-tokyo.ac.jp

## Abstract

This paper proposes a dual variational autoencoder (DualVAE), a framework for generating images corresponding to multiclass labels. Recent research on conditional generative models, such as the Conditional VAE, exhibit image transfer by changing labels. However, when the dimension of multiclass labels is large, these models cannot change images corresponding to labels, because learning multiple distributions of the corresponding class is necessary to transfer an image. This leads to the lack of training data. Therefore, instead of conditioning with labels, we condition with latent vectors that include label information. DualVAE divides one distribution of the latent space by linear decision boundaries using labels. Consequently, DualVAE can easily transfer an image by moving a latent vector toward a decision boundary and is robust to the missing values of multiclass labels. To evaluate our proposed method, we introduce a conditional inception score (CIS) for measuring how much an image changes to the target class. We evaluate the images transferred by DualVAE using the CIS in CelebA datasets and demonstrate state-of-the-art performance in a multiclass setting.

## 1 Introduction

Recent conditional generative models have shown remarkable success in generating and transferring images. Specifically, a conditional variational autoencoder (CVAE) (Kingma et al., 2014) can generate conditional images by learning the latent space $\mathcal{Z}$ that corresponds to multiclass labels. In addition, StarGAN (Choi et al., 2017) and FaderNetworks (Lample et al., 2017) can generate images corresponding to multiple domains by conditioning with domains such as attributes.

However, when the dimension of the multiclass is increased, these models cannot transfer the images corresponding to one arbitrary domain (an element of a multiclass label). The possible reasons are the following. For simplicity, we consider a binary multiclass classification. To transfer an image of a certain class, it is necessary to learn the distributions of the corresponding class. That is, assuming that the number of classes in the multiclass is N, conditional models need to create $2^N$ distributions. However, when N is large, training is difficult as $\mathcal{O}(2^N)$ training samples will be required.

Hence, instead of conditioning with labels, we propose DualVAE, which conditions with latent vectors that include label information. DualVAE divides one distribution of the latent space by N linear decision boundaries which need to learn only $\mathcal{O}(N)$ parameters by adding another decoder $p_w(\mathbf{y}|\mathbf{z})$ to a variational autoencoder (VAE) (Kingma & Welling, 2013). DualVAE assumes that a label is a linear combination of vectors of the latent space and the dual latent space. There are two advantages to the DualVAE decoder $p_w(\mathbf{y}|\mathbf{z})$ being a *linear* model. First, DualVAE can easily transfer an image by moving a latent vector toward a decision boundary. Next, DualVAE is robust to the missing values of multiclass labels.

In addition to this method, we propose the conditional inception score (CIS), a new metric for conditional transferred images. Although the evaluation methods often used in the generation models are the Inception Score (IS) (Salimans et al., 2016) and the Fréchet Inception Distance (Heusel et al., 2017), they are used for evaluating the diversity of images and not suitable for evaluating transferred images conditioned with domains such as attributes or classes. Therefore, we propose a new metric to evaluate two properties: the first property pertains to whether images in one domain are transferred properly to images in another domain; the second property pertains to whether images in one domain

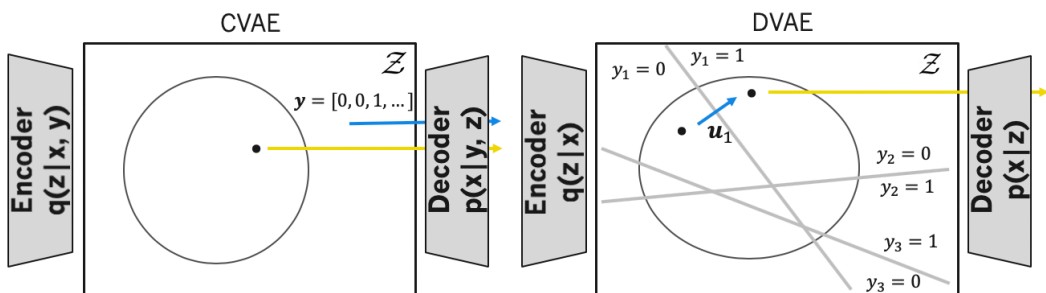

Figure 1: Conditional VAE learns $2^n$ distributions for each binary multiclass label when the number of class is n. DualVAE learns n decision boundaries for dividing a distribution of latent space. $\mathbf{u}_1$ is a parameter of a decision boundary, which we call a dual vector.

transferred to images in another domain can preserve the original properties. By using the CIS, we compare DualVAE with other methods that can perform image-to-image translations for multiple domains.

In summary, the contributions from this study are as follows: 1) We introduce DualVAE, a method for transferring images corresponding to multiclass labels and demonstrate that images can be transferred quantitatively and qualitatively. 2) We propose the CIS, a new metric that can evaluate transferred images corresponding to multiclass labels.

## 2 RELATED WORK

**Conditional model** Several studies have been conducted to generate or transfer images conditioned with labels. For example, conditional VAE (Kingma et al., 2014) is an extension of a VAE (Kingma & Welling, 2013) where latent variables $z$ are inferred using image $x$ and label $y$, and image $x$ is reconstructed with $y,z$. Further, a CGAN (Mirza & Osindero, 2014) is a conditional model using a GAN, where a noise $z$ and a class label $y$ are input to the generator, and learning is performed similarly to the GAN using image $x$ corresponding to class label $y$. FaderNetworks (Lample et al., 2017) learns latent variables from which label information is eliminated by using adversarial learning and assigns attributes to images by providing labels to the decoder. Furthermore, StarGAN (Choi et al., 2017), a method of domain transfer, had succeeded in outputting a beautiful image corresponding to an attribute by conditioning with a domain (attribute). However, all these methods are models conditioned with labels; therefore, as the dimension of the labels becomes larger, the number of training samples becomes insufficient.

**Connection to the Information Bottleneck** As with DualVAE, there are several papers related to finding a latent variable z that predicts label y. For example, Information Bottleneck (IB) (Tishby et al., 2000) is a method for obtaining a latent expression z that solves task y. IB is a method which leaves the latent information z for solving the task y by maximizing the mutual information amount I(Z; Y). At the same time, extra information about input x is discarded by minimizing I(Z; X). Variational Information Bottleneck (VIB) (Alemi et al., 2016) succeeded in parameterizing the IB with a neural network, by performing a variational approximation. VIB can also be considered as a kind of extension of VAE. VAE minimizes the mutual information I(Z; i) between individual data i and latent variable z while maximizing I(Z; X). DualVAE can be regarded as a framework of VIB as well, and it minimizes I(Z; i) while maximizing I(Z; Y) and I(Z; X).

**Connection to the Probabilistic Matrix Factorization** We can also regard DualVAE as a probabilistic matrix factorization (PMF) (Mnih & Salakhutdinov, 2008) extended to a generative model. A PMF is used in several application areas, primarily in collaborative filtering, which is a typical recommendation algorithm. It can predict missing ratings of users by assuming that the user's ratings are modeled by a linear combination of the item and user latent factors. Similarly, we experimentally show that DualVAE is also robust to missing labels.

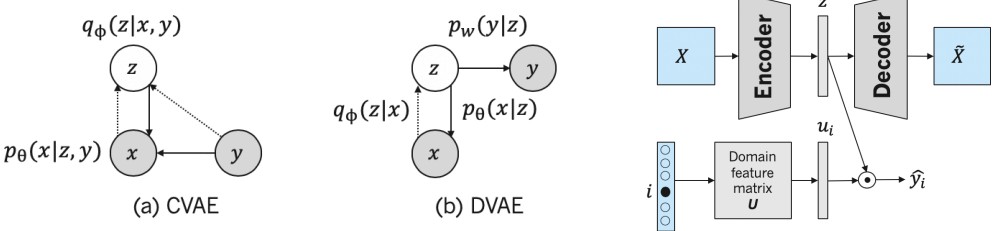

Figure 2: **Left**: Graphical models of the probabilistic models of Conditional VAE and DualVAE. The gray and white circles indicate the observed variables and latent variables, respectively. Arrows between the symbols indicate probabilistic dependency. **Right:** The network structure of DualVAE. The predicted label is structured as the inner product of latent $\mathbf{z}$ and dual vector $\mathbf{u}_i$.

## 3 DUALVAE

We devised DualVAE by adding a decoder $p_w(y|\mathbf{z}) = p(y|\mathbf{z}, \mathbf{u})$ to the VAE to learn the decision boundaries between classes. Here, $\mathbf{z}$ is a vector of the latent space $\mathcal{Z}$, $\mathbf{u}$ is a vector of the dual latent space $\mathcal{Z}^*$ and $y$ is a label. Unlike the CVAE, this model does not require label $\mathbf{y}$ at the time of inference of $\mathbf{z}$ corresponding to $\mathbf{x}$, and the difference is shown in Figure 2. The objective function of the VAE is as follows:

$$\mathcal{L}(\mathbf{x}) = \mathbb{E}_{q_\phi(\mathbf{z}|\mathbf{x})}[\log p_\theta(\mathbf{x}|\mathbf{z})] - D_{\mathrm{KL}}\left(q_\phi(\mathbf{z}|\mathbf{x})||p(\mathbf{z})\right), \qquad (1)$$

where $\phi, \theta$ are parameters of the encoder and decoder of the VAE, respectively. The lower bound of DualVAE is as follows:

$$
\begin{aligned}
\log p_\theta(\mathbf{x}, \mathbf{y}) &\geq \mathbb{E}_{q_\phi(\mathbf{z}|\mathbf{x})}\left[\log \frac{p_\theta(\mathbf{x}, \mathbf{y}, \mathbf{z})}{q_\phi(\mathbf{z}|\mathbf{x})}\right] \\
&= \mathbb{E}_{q_\phi(\mathbf{z}|\mathbf{x})}[\log p_\theta(\mathbf{x}|\mathbf{z}) + \log p(\mathbf{z}) - \log q_\phi(\mathbf{z}|\mathbf{x}) + \log p_w(\mathbf{y}|\mathbf{z})] \\
&= \mathcal{L}(\mathbf{x}) + \mathbb{E}_{q_\phi(\mathbf{z}|\mathbf{x})}\left[\log p_w(\mathbf{y}|\mathbf{z})\right], \qquad (2)
\end{aligned}
$$

where $p_w(\mathbf{y}|\mathbf{z}) = Bern(\mathbf{y}|\sigma(\mathbf{U}\mathbf{z}))$. Here, U is a domain feature matrix whose row vector is a dual vector $\mathbf{u}$ and Bern is a Bernoulli distribution. As you can see from Equation 2, the objective function of DualVAE is the objective function of the VAE plus the expectation of log-likelihood of $p_w(\mathbf{y}|\mathbf{z})$ Specifically, training is performed such that the inner product of $\mathbf{z}_j \in \mathcal{Z}$ and $\mathbf{u}_i \in \mathcal{Z}^*$ predicts the label $y_{ij}$ where j is the index of a sample and i is the index of a domain. At the same time, we find the values of $\theta$ and $\phi$ that maximize the lower bound in Equation 1.

We transfer the images on domain i by performing the following operation. We calculated the following vector $\mathbf{w}_i$:

$$\mathbf{w}_i = \mathbf{z} + \lambda \boldsymbol{u}_i, \qquad (3)$$

where $\lambda (\in \mathcal{R})$ is a parameter. Image transfer can be demonstrated by changing $\lambda$ and decoding $\mathbf{w}_i$. Equation 3 corresponds to moving a latent vector toward a decision boundary.

---

**Algorithm 1** DualVAE
---
**Require:** images $(x_j)_{j=1}^m$, batch size $M$, indicator function $I_{ij}$ VAE/encoder optimizers: $g$, $g_e$, hyper parameter $\alpha$, and the label matrix $Y = (y_{ij})$.
    Initialize encoder parameter, decoder parameter and dual vector: $\theta, \phi, U = (u_i)_{i=1}^n$
    **repeat**
        Randomly select batch $(x_j)_{j\in\mathcal{B}}$ of size $M$
        Sample $z_j \sim q_\phi(z_j|x_j) \; \forall j \in \mathcal{B}$
        $\phi, \theta \leftarrow g(\nabla_{\phi,\theta} \sum_{j\in\mathcal{B}}[\log p_\theta(x_j|z_j) - D_{\mathrm{KL}}(q_\phi(z_j|x_j)||p(z))])$
        $\phi, U \leftarrow g_e(\nabla_{\phi,U} \sum_i \sum_{j\in\mathcal{B}} \alpha I_{ij} \log p(y_{ij}|z_j, u_i))$
    **until** convergence of parameters $\theta, \phi, U$

---

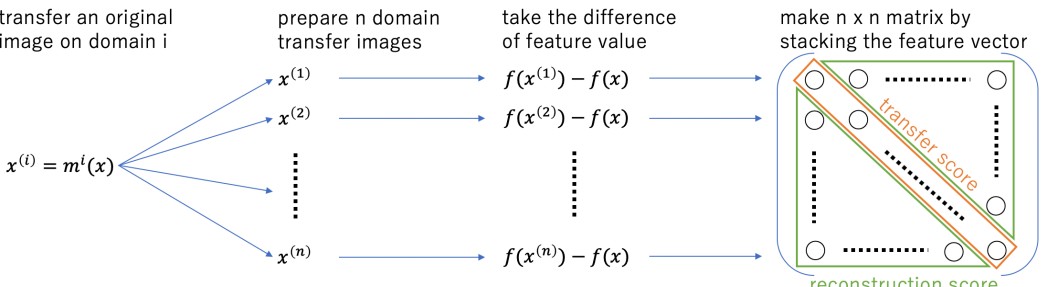

Figure 3: Computation flow of the transfer score and the reconstruction score for CIS. $x$ is an original image. $x^{(i)}$ denotes a transferred image on domain i.

## 4   CIS

Although IS (Salimans et al., 2016) is a score for measuring generated images, it can only measure the diversity of the images, and cannot be used for evaluating the domain transfer of the images. Therefore, we proposed using a CIS, a score for evaluating the transformation of images into multiclass target domains.

The CIS is a scalar value calculated from the sum of two elements. The first is whether the domain transfer of the original image has been successful (transfer score), and the second is whether the features other than the transferred domain are retained (reconstruction score). The computation flow of these scores can be found in Figure 3.

We calculated the CIS using Algorithm 2. First, we assumed that the number of domains is n and the domain that each image belonged to was known. We finetuned Inception-v3 (Szegedy et al., 2016) using train images as inputs and domains as outputs. To enable the model to classify the images with the domains, we replaced the last layer of the model with a new layer that had n outputs. Next, we transferred test images into n domain images and loaded the transferred images into the pretrained Inception-v3. Through this process, we obtained an $n \times n$ matrix for every original image because one image was transferred into n domain images and each domain image was mapped to an n-dimension vector. We subsequently mapped the original image into an n-dimension vector using Inception-v3 and subtracted this vector from each row of the $n \times n$ matrix. We named this matrix M. The key points are the following: (1) the diagonal elements of M should be large because the specified domain should be changed significantly, and (2) the off-diagonal elements of M should be small because the transferred images should preserve the original features.

---

**Algorithm 2** Conditional Inception Score (CIS)

---

**Require:** observation $x \in \mathcal{X}$, Inception-v3 $f : \mathcal{X} \to \mathcal{R}^n$, domain transfer model $m$.
   **for** $i = 1 \ldots n$ **do**
      $x^{(i)} \leftarrow m^i(x)$
      $M_i \leftarrow f(x^{(i)}) - f(x)$
   **end for**
   ts $\leftarrow$ avg(diag($M$))
   rs $\leftarrow -$avg(abs(notdiag($M$)))
   CIS $\leftarrow$ ts+rs

---

In the algorithm, abs denotes taking the absolute value, diag denotes taking the diagonal elements of the matrix, notdiag denotes taking the nondiagonal elements, avg denotes taking the mean of the multiclass values. $x$ is an original image, and $x^{(i)}$ denotes a transferred image on domain i.

## 5 EXPERIMENT

We performed a standard image transfer task with the 40 attributes in the CelebA (Liu et al., 2015) dataset, which comprises approximately 200,000 images of faces of celebrities.

**Comparison of DualVAE and several models** DualVAE was compared with several models capable of performing image-to-image translations for multiclass labels using a single model. In each model, we calculated the CIS several times when applying Algorithm 2 on 160 CelebA test images; subsequently, the average and standard deviation were obtained. DualVAE obtained a higher CIS than the other models and the results are shown in Table 1 and Figure 4.

Table 1: Average CISs for three conditional models, which demonstrates DualVAE outperforms several models based on the CIS.

| Method | 20 domains | 40 domains |
|---|---|---|
| CVAE (Kingma et al., 2014) | -0.112±0.007 | -0.152±0.006 |
| FaderNetworks (Lample et al., 2017) | 0.075±0.008 | -0.002±0.003 |
| StarGAN (Choi et al., 2017) | 0.068±0.188 | 0.050±0.032 |
| DualVAE | **0.163**±0.025 | **0.140**±0.020 |

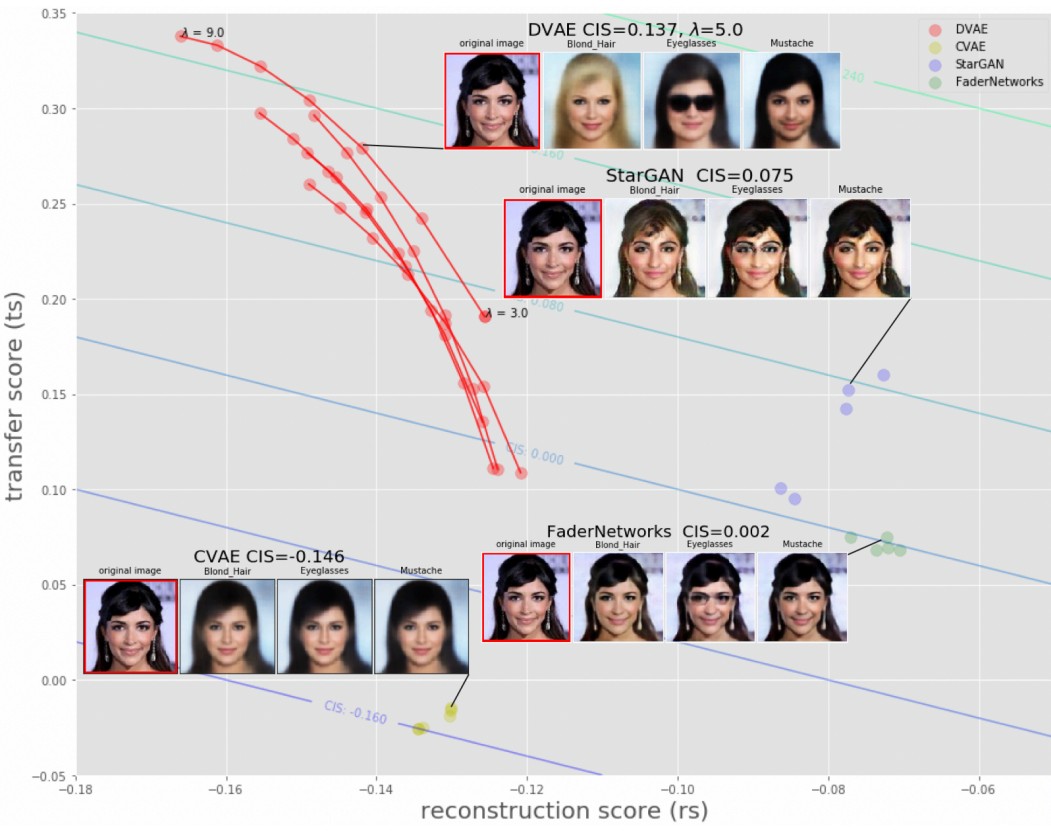

Figure 4: Scatter plot of DualVAE, FaderNetworks, StarGAN and CVAE. Different color denotes different models. All of the transferred images are in Figure 5.

**Visualization of transferred images** We visualized transferred images of the 40 attributes by the proposed method and other models in Figure 5. Although StarGAN and FaderNetworks retained the characteristics of the original image considerably, it was qualitatively understood that domain transfer was poor.

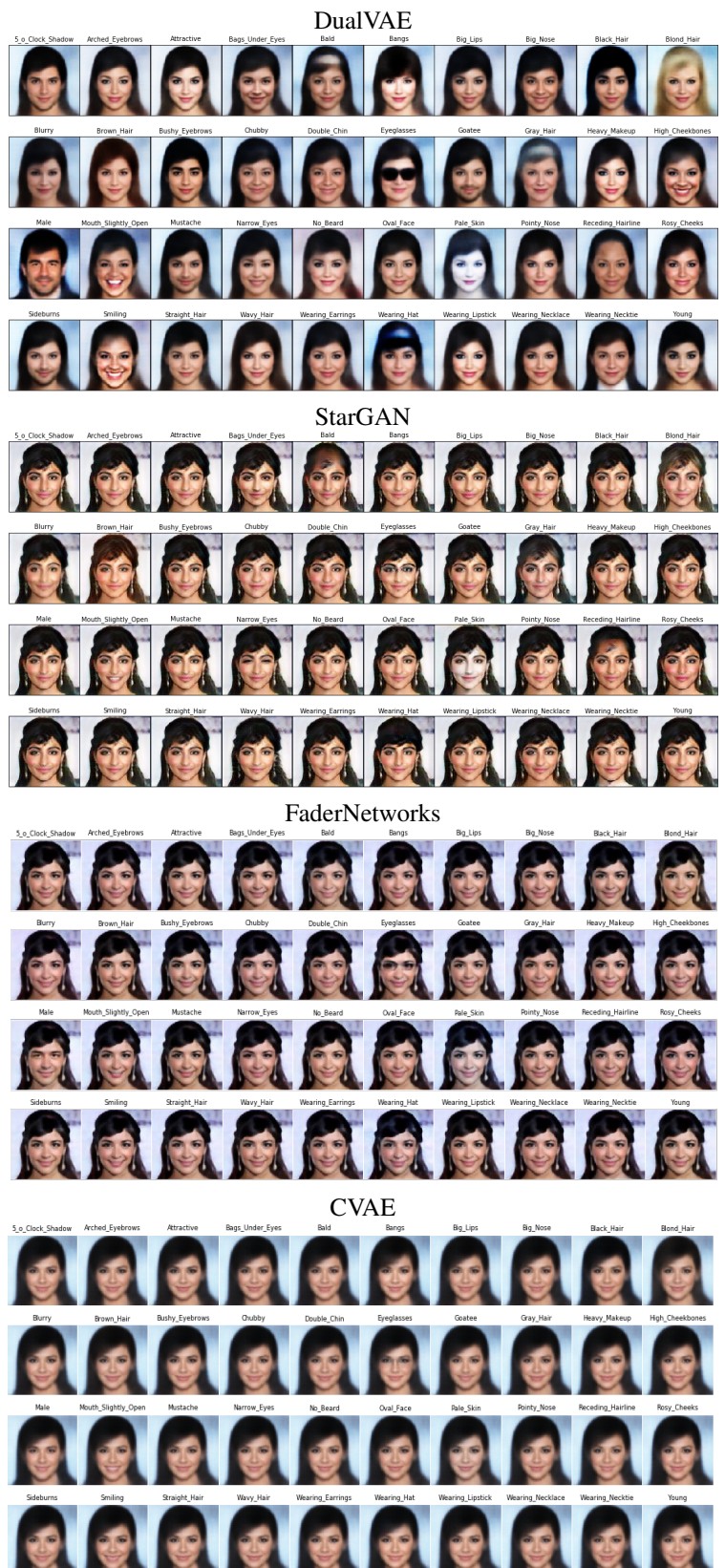

Figure 5: DualVAE transferred images of the 40 attributes well; however, StarGAN, FaderNetworks and CVAE did not transform the images well.

**Robustness to sparsity**    To demonstrate experimentally that DualVAE is robust to the missing values of multiclass labels, the following steps were performed. We calculated the rs and ts values when applying Algorithm 2 on 160 CelebA test images and plotted the figure below when we changed the missing ratio of CelebA's domain labels and the $\lambda$ in Equation 3.

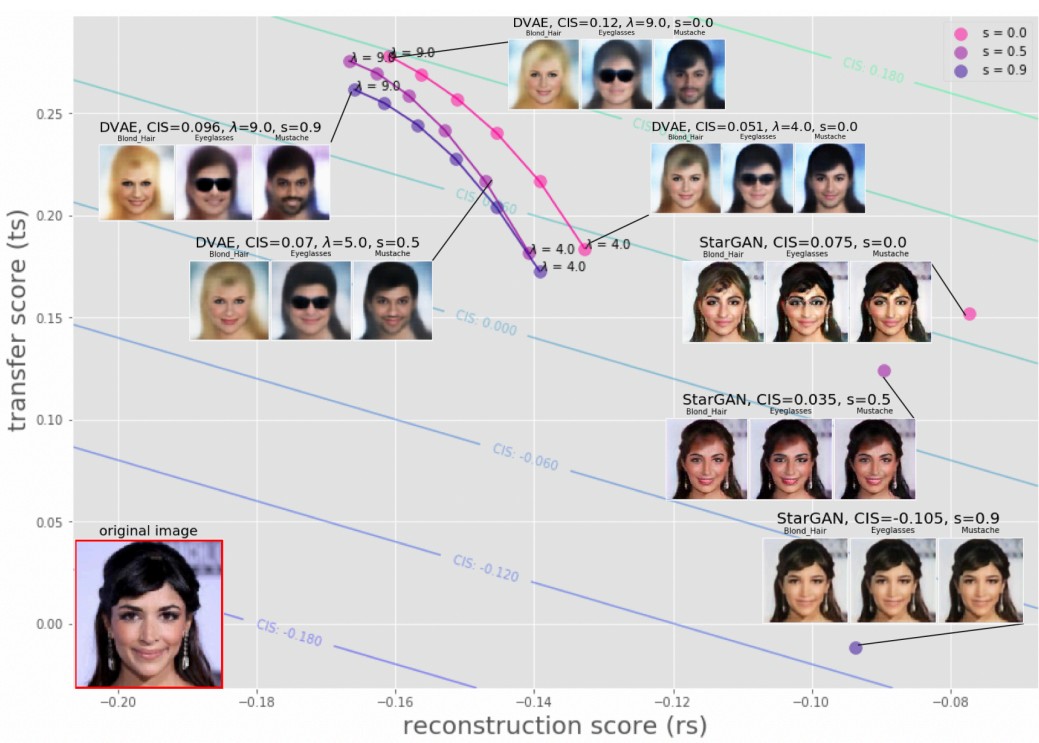

Figure 6: Scatter plot of the missing ratio of CelebA's label and CIS of DualVAE. Variable s is the missing ratio. The original image is shown on the bottom left of the figure. The attributes of the original images are transformed into blond hair, eyeglasses, and a mustache. The vertical axis is the ts value of Algorithm 2 and the horizontal axis is the rs value of Algorithm 2. CIS grows in the upper right corner.

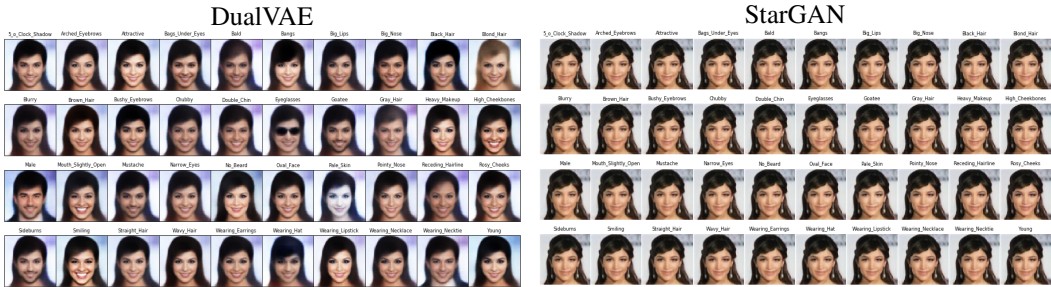

(a) $s = 0.9$. Image transfer was still well-conducted.

(b) $s = 0.9$. All identical images were generated, and image transfer was not conducted properly.

As shown in Figure 6, DualVAE is robust in terms of the sparseness of domain labels, and the CIS does not decrease even when 90% of the labels are missing. Meanwhile, we found that StarGAN is not as robust as DualVAE with respect to sparseness. When 90% of the domain labels are missing, StarGAN cannot learn at all and generates identical images.

## 6 CONCLUSION AND FUTURE WORK

We proposed DualVAE, a simple framework for generating and transferring images corresponding to multiclass labels. Further, we introduced the CIS, a new metric for measuring how much of an image corresponding to the change of labels could be generated. The decoder of DualVAE was a simple linear model in this study; however, we would like to test more complex models in the future.

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
