# OpenReview forum: "DUAL SPACE LEARNING WITH VARIATIONAL AUTOENCODERS"
_ICLR.cc/2019/Workshop/DeepGenStruct — DeepGenStruct 2019_

### Official Review · AnonReviewer1 · 2019-04-15
**The paper is not well-written and the proposed metric is not justified**

**Rating:** 2
**Confidence:** 2

**Review:**

This paper proposes a dual variational autoencoder(DualVAE) model for generating images with multi labels. Compared to the existing methods like conditional VAE, the difference is the introduction of a dual vector to the model, which is a parameter of a decision boundary. The author claims that such a change can avoid the exponential dependency of the number of class.

The second contribution of the paper is to propose a new metric to evaluate the transferred images corresponding to multiclass labels, i.e. Conditional Inception Score (CIS).

Empirical studies are conducted on the CelebA dataset. DualVAE achieves better CIS scores than three existing methods.

The paper is not well-written and hard to follow. Some places are not precisely stated, e.g. “exhibit image transfer by changing labels” in the abstract. The experiment is not convincing, as it only compares different methods on the proposed metric. How do you justify CIS is an appropriate metric?

---

### Official Review · AnonReviewer2 · 2019-04-17
**looks good. I mainly have concerns about the inference procedure.**

**Rating:** 3
**Confidence:** 2

**Review:**

This work describes a dual variational autoencoder for image style transfer. Different from CVAE, the conditional assumption here is that a label is dependent on the latent representation of the image. The other contribution is a metric for evaluating conditionally transferred images.

The paper is generally well-written and explained in most parts. Motivations are clear. The experimental results look good.

However, I am not very convinced about the inference procedure of the model, as shown in Equation (3). This looks artificial, since you are feeding the linear combination of the latent and dual code (i.e., the label space) into the decoder. I see no reason why this will give good transferred image, since the decoder is trained with different input. I personally think you can have an inference network which takes the latent code and the label to generate the image. Did I miss something?

---

### Decision · Program_Chairs · 2019-04-19
**Acceptance Decision**

**Decision:**

Accept

**Comment:**

This paper proposes a VAE for images with multiple labels and a new evaluation metric called Conditional Inception Score (CIS) that measures the influence of the target class to the image. The experiments are reasonable.